# Leading the Follower: Learning Persuasive Agents in Social Deduction Games

## Abstract

Large language model (LLM) agents have shown remarkable progress in social deduction games (SDGs). However, existing approaches primarily focus on information processing and strategy selection, overlooking the significance of persuasive communication in influencing other players' beliefs and responses. In SDGs, success depends not only on making correct deductions but on convincing others to response in alignment with one's intent. To address this limitation, we formalize turn-based dialogue in SDGs as a Stackelberg competition, where the current player acts as the leader who strategically influences the follower's response. Building on this theoretical foundation, we propose a reinforcement learning framework that trains agents to optimize utterances for persuasive impact. Through comprehensive experiments across three diverse SDGs, we demonstrate that our agents significantly outperform baselines. This work represents a significant step toward developing AI agents capable of strategic social influence, with implications extending to scenarios requiring persuasive communication.

## 1 Introduction

Large language model (LLM) agents have demonstrated remarkable capabilities across diverse domains, from computer desktop interactions (Nayak et al., 2025; Wang & Liu, 2025; Wang et al., 2025) to game environments like StarCraft (Ma et al., 2024; Shao et al., 2024a) and Minecraft (Wang et al., 2024a; Fu et al., 2025; Chai et al., 2025). These successes showcase the potential of LLMs in sequential decision-making and complex reasoning tasks. However, most existing applications involve agents interacting with deterministic, rule-based environments where feedback follows consistent patterns and information is inherently truthful. In contrast, real-world human interactions involve uncertainty, deception, and strategic communication, presenting fundamentally different challenges for AI agents.

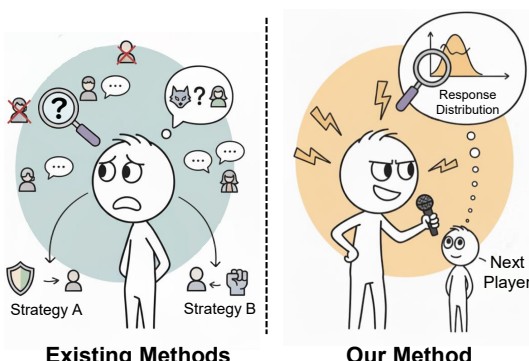

Figure 1: **Methodological paradigms.** Existing methods primarily focus on processing environmental information (such as identifying other players' roles) and selecting strategies based on this information. In contrast, our method measures the next player's response distribution and optimizes utterances specifically for persuasive impact on subsequent player responses.

Social deduction games (SDGs) such as Werewolf (Zhang et al., 2025; Xu et al., 2023; 2024; Wu et al., 2024), Avalon (Light et al., 2023; Wang et al., 2023; Shi et al., 2023; Lan et al., 2024), and ONUW (Jin et al., 2024; Lai et al., 2023) provide ideal testbeds for developing agents that can handle these complexities. Recent work on LLM agents for SDGs has made notable progress. Approaches include using prompt engineering to perform reflection (Lan et al., 2024; Wang et al., 2024b), generating code for enhanced deduction (Light et al., 2025) and reinforcement learning (RL) for strategy selection (Xu et al., 2024; Jin et al., 2024; Xu et al., 2025).

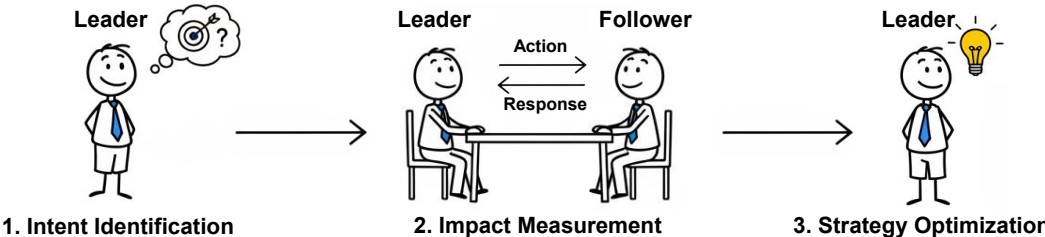

Figure 2: **Stackelberg optimization process.** First, the leader identifies their strategic intent by analyzing the current situation. Then, the leader measures the follower's response distribution to different leader actions. Finally, the leader optimizes their strategy to maximize their utility given the follower's response distribution.

However, these methods largely adapt techniques from general task domains without fully leveraging the unique persuasive dynamics of social games. They typically concentrate on deducing other players' roles and making strategic choices, neglecting the critical ability to influence other players' beliefs and responses through persuasive communication.

The core challenge in SDGs extends beyond processing complex information and selecting optimal strategies. Success fundamentally depends on persuasive communication, which is the ability to shape how other players think and response. In Werewolf, for instance, a villager must not only identify werewolves but craft utterances that convince other players to vote accordingly. This persuasive dimension, central to both gameplay success and real-world human interaction, remains largely unaddressed in current research.

As shown in Figure 1, we address this gap by explicitly modeling and optimizing persuasive communication in SDGs. We formalize turn-based dialogue as a Stackelberg competition, a sequential game where one player (the leader) takes an action first, while the other player (the follower) responds accordingly. Figure 2 illustrates the key insight that in a Stackelberg competition, if the leader sufficiently understands how the follower will response, they can maximize their utility subject to the follower's response distribution as a constraint. In our context, at each turn in our collected self-play dataset, the current player acts as the leader who optimizes their utterances by measuring the next player's response distribution. This formulation captures the strategic nature of persuasion and directly guides our training framework design, which is crafting utterances that proactively steer subsequent conversations toward desired outcomes.

Building on this theoretical foundation, we propose an RL framework for training persuasive agents. Our method fine-tunes an LLM on our collected self-play dataset to refine base utterances into persuasive ones. The persuasive impact is measured by calculating how much an utterance shifts the probability distribution of the follower's responses toward desired outcomes. Since our Stackelberg optimization requires comparing different utterances based on their potential to elicit desired follower responses, we use GRPO (Shao et al., 2024b) to fine-tune the LLM, which efficiently computes relative advantages without requiring an explicit critic model. Through this approach, our agents learn to craft utterances that maximize persuasive impact.

We validate our approach through extensive experiments across three SDGs: Werewolf, Avalon, and ONUW. Results demonstrate that our agents significantly outperform baseline methods, achieving higher win rates by effectively guiding conversations and influencing other players' behaviors. The improvements are evident both in scenarios requiring trust-building and coordination as well as in deceptive roles.

Our contributions can be summarized as follows:

- We formulate turn-based dialogue in SDGs as a Stackelberg competition, providing a systematic and theoretical foundation for analyzing and optimizing persuasive communication.

- We propose a training framework that optimizes utterances to directly influence subsequent players' responses, enabling agents to proactively steer conversational flow toward desired outcomes while avoiding undesired ones.

- We demonstrate effectiveness and generalizability through comprehensive experiments on three SDGs, showing that our agents achieve superior performance through more persuasive communication.

## 2 RELATED WORK

### 2.1 SOCIAL DEDUCTION GAME AGENTS

Social deduction games (SDGs) are multiplayer games where players are assigned hidden roles and must use communication, deduction, and deception to achieve their objectives. Early works on SDG agents (Osawa et al., 2014; Wang & Kaneko, 2018) often rely on rule-based systems or predefined communication templates, limiting their adaptability and expressive power. More recently, LLMs have become the backbone of agents that can engage in free-form dialogue. Some works have demonstrated the effectiveness of prompt-based methods. For instance, Xu et al. (2023) develop Werewolf agents using information retrieval and experience reflection, ReCon (Wang et al., 2024b) prompts agents to play Avalon by reasoning from both their own and their opponents' perspectives, and Strategist (Light et al., 2025) generates strategies as code and uses tree search for selection.

To address the limited exploration of purely prompt-based agents, researchers have begun integrating reinforcement learning (RL). Wu et al. (2024) train a Thinker module to select an action from a predefined action space. Similarly, SLA (Xu et al., 2024) learns a policy to select from a set of candidate actions generated by an LLM, and LSPO (Xu et al., 2025) defines a finite strategy space by clustering and trains a policy to select among them. These methods typically reduce the rich space of natural language communication to classification problems. They learn to select from limited candidates rather than optimizing utterances in the natural language domain. Our work diverges by using RL to refine utterances directly within the continuous space of natural language, enabling more nuanced and flexible persuasive communication.

### 2.2 REINFORCEMENT LEARNING FOR LLMS

Reinforcement learning (RL) has been widely adopted to improve LLMs by optimizing them toward specific objectives through reward signals. Common approaches include PPO (Ouyang et al., 2022; OpenAI et al., 2024), which trains a reward model from human preferences and uses it to provide feedback during policy optimization, and DPO (Rafailov et al., 2023), which directly optimizes the policy by maximizing the log likelihood ratio between chosen and rejected responses from preference pairs. These methods have proven effective across diverse applications, from preference alignment (Liang et al., 2024; Yang et al., 2024; Khanov et al., 2024) to sequential decision-making in interactive environments (Szot et al., 2024; Yao et al., 2022; Verma et al., 2022). To reduce the reliance on human-annotated preference data, algorithms like Group Relative Policy Optimization (GRPO) (Shao et al., 2024b) have been developed. GRPO leverages the distribution of rewards within each training batch to construct relative preference signals, which eliminate the need for separate critic models and explicit preference pairs. In this paper, we employ GRPO to train agents to refine their utterances for persuasive communication in SDGs.

### 2.3 GAME-THEORETIC MODELS OF COMMUNICATION

Game theory provides a principled framework for analyzing strategic interactions in multi-agent systems. SDGs are commonly modeled as extensive-form Bayesian games (Fudenberg & Tirole, 1991), capturing both the sequential nature of actions and the uncertainty arising from hidden information. Traditional game-theoretic approaches typically seek equilibrium solutions, such as Perfect Bayesian Equilibrium (PBE), where players' strategies and beliefs form a stable configuration in which no individual player benefits from changing their strategy while others keep theirs fixed. These theoretical foundations have enabled the development of sophisticated game agents. Deep-Role (Serrino et al., 2019) combines deductive reasoning with Counterfactual Regret Minimization (CFR) to achieve superior performance in non-communicative Avalon, while Cicero (Bakhtin et al., 2022) extends this success to language-based games by simplifying the natural language space into a finite decision space for strategic selection.

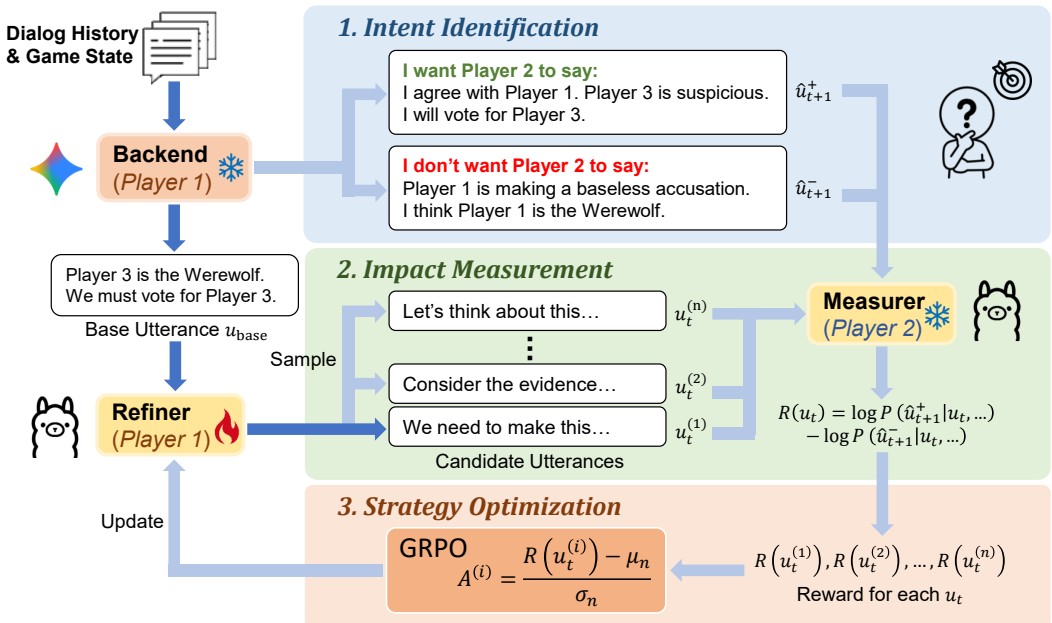

Figure 3: **The training framework of our agent.** Dark blue arrows indicate the inference pipeline, while light blue arrows represent additional processes during training. In this instance, Player 1 acts as the leader while Player 2 acts as the follower. The backend LLM identifies desired and undesired target responses, then generates a base utterance $u_{\text{base}}$. The Refiner enhances $u_{\text{base}}$ for maximum persuasive impact. The Measurer computes rewards by measuring how different refined utterances $u_t$ affect the probabilities of generating $\hat{u}^+_{t+1}$ and $\hat{u}^-_{t+1}$. Multiple utterances $u_t$ are sampled for group relative advantage calculation during training, while only one is generated during inference. The backend uses an API-based LLM, while the Refiner and Measurer are two copies of the same open-source LLM, with the Measurer's parameters frozen.

However, computing global equilibria becomes computationally intractable when the action space includes free-form natural language. These approaches also overlook the turn-by-turn persuasive dynamics that characterize human communication. To address these limitations, we adopt the Stackelberg competition model, which treats each speaking turn as a leader's opportunity to influence the follower's response. This local optimization approach captures the strategic essence of persuasion while remaining computationally feasible, providing both theoretical grounding and practical applicability to language-based interactions in SDGs.

## 3 METHOD

### 3.1 PRELIMINARY

The Stackelberg competition is a strategic game model where one player (the leader) takes an action first, while the other player (the follower) responds accordingly. This sequential structure creates an information advantage that the leader can exploit through strategic foresight. In real-world scenarios, the follower's response may not be deterministic due to bounded rationality, incomplete information, or inherent randomness in decision-making. To handle this uncertainty, the leader must work with probabilistic response distributions. If the leader sufficiently understands how the follower will respond to different actions, they can maximize their utility by selecting actions that optimize their expected outcomes given the follower's response distribution as a constraint.

In turn-based dialogue scenarios like SDGs, this framework naturally applies where the current player serves as the leader who can influence subsequent conversation flow through strategic communication. Therefore, we model the interaction between consecutive players as a two-player Stack-

elberg competition, where the current player acts as the leader, and the next player acts as the follower. We provide further analysis for this in Appendix B.

During training, each speaking turn constitutes a training instance. As shown in Figure 3, our training framework involves three key steps: (1) **Intent Identification**: the leader identifies their strategic intent based on the current situation. (2) **Impact Measurement**: the leader measures the potential impact of different utterances on the follower's response. (3) **Strategy Optimization**: the leader optimizes their strategy to maximize expected utility given the follower's response distribution.

## 3.2 INTENT IDENTIFICATION

We formalize an SDG as a multi-agent sequential game with a set of players, where each player $p$ is assigned a hidden role $r$ at initialization. The game proceeds through several rounds, with each round consisting of a discussion phase and an action phase (e.g., voting, elimination).

During the discussion phase, players take turns making utterances. For player $p_t$ speaking at turn $t$, their utterance generation depends on four key inputs: the game rules $\mathcal{R}$ that specify game mechanics and win conditions; the current game state $G_t$ that includes publicly observable information such as eliminated players, revealed roles, and voting outcomes; the dialogue history $D_t = \{u_1, u_2, \ldots, u_{t-1}\}$ that contains all previous utterances; and their hidden role $r_t$.

At turn $t$, the current player $p_t$ acts as the leader, while the next player $p_{t+1}$ acts as the follower. Given the current context $(\mathcal{R}, G_t, D_t, r_t)$, the leader first identifies its persuasive intent: a desired response $\hat{u}_{t+1}^+$ that would be most advantageous if spoken by $p_{t+1}$, and an undesired response $\hat{u}_{t+1}^-$ that would be most disadvantageous. We employ the agent's backend LLM to analyze the situation and identify $\hat{u}_{t+1}^+$ and $\hat{u}_{t+1}^-$:

$$(\hat{u}_{t+1}^+, \hat{u}_{t+1}^-) = f_{\text{identify}}(\mathcal{R}, G_t, D_t, r_t). \tag{1}$$

## 3.3 IMPACT MEASUREMENT

Modern API-based LLMs (e.g., GPT-5, Gemini-2.5) typically demonstrate superior reasoning and generation capabilities compared to open-source alternatives. However, their closed-source nature prevents direct fine-tuning for specific objectives. To leverage the strengths of both paradigms, we adopt a two-stage approach. We first use an API-based LLM as our backend LLM for generating base utterances, then fine-tune an open-source LLM as an auxiliary component to refine this utterance for maximum persuasive impact.

Specifically, given a base utterance $u_{\text{base}}$ generated by the agent's backend LLM:

$$u_{\text{base}} = f_{\text{base}}(\mathcal{R}, G_t, D_t, r_t). \tag{2}$$

We fine-tune the Refiner $\pi_\theta$ that transforms $u_{\text{base}}$ into a more persuasive version:

$$u_t \sim \pi_\theta(\cdot | u_{\text{base}}, \mathcal{R}, G_t, D_t, r_t). \tag{3}$$

During inference, $u_t$ is used as the agent's final utterance, while during training, we sample a group of $n$ candidate utterances from the Refiner:

$$\{u_t^{(1)}, u_t^{(2)}, \ldots, u_t^{(n)}\} \sim \pi_\theta(\cdot | u_{\text{base}}, \mathcal{R}, G_t, D_t, r_t). \tag{4}$$

For each candidate utterance $u_t^{(i)}$, we measure its persuasive impact using a reward function that quantifies the shift in follower response probabilities:

$$\begin{aligned} R(u_t^{(i)}) = &\log P_{\mathcal{F}}(\hat{u}_{t+1}^+ | \mathcal{R}, G_{t+1}, D_t \cup \{u_t^{(i)}\}, r_{t+1}) \\ &- \log P_{\mathcal{F}}(\hat{u}_{t+1}^- | \mathcal{R}, G_{t+1}, D_t \cup \{u_t^{(i)}\}, r_{t+1}), \end{aligned} \tag{5}$$

where $P_{\mathcal{F}}$ represents the follower's probability of generating a response.

Since the follower's backend is also an API-based LLM that we cannot access for probability computation, we use an open-source LLM as the Measurer to simulate the follower's response patterns. The Measurer and Refiner are two copies of the same LLM, with the Measurer's parameters frozen.

The Measurer only serves to provide rewards for the Refiner's outputs rather than actually participating as the next player in the game.

To compute $P_\mathcal{F}$, we construct a prompt with the game context and the dialogue history including $u_t$, then append the target response $\hat{u}_{t+1}$. The probability is calculated as:

$$\log P_\mathcal{F}(\hat{u}_{t+1}|\mathcal{R}, G_t, D_t \cup \{u_t\}, r_{t+1}) = \sum_{i=1}^{|\hat{u}_{t+1}|} \log p(w_i|w_{<i}, \mathcal{R}, G_t, D_t \cup \{u_t\}, r_{t+1}), \quad (6)$$

where $w_i$ denotes the $i$-th token in the target response $\hat{u}_{t+1}$, and $p$ is the probability assigned to each token by the Measurer through a single autoregressive forward pass. We use log probabilities to prevent numerical underflow and ensure stable optimization. Since the game state typically changes during the action phase and remains constant during the discussion phase, we approximate $G_{t+1}$ with $G_t$. While the next player's hidden role $r_{t+1}$ is unobservable to player $p_t$ during evaluation, we leverage full information during training to create a more accurate reward.

### 3.4 STRATEGY OPTIMIZATION

We optimize the Refiner $\pi_\theta$ using Group Relative Policy Optimization (GRPO) (Shao et al., 2024b), which does not require human-annotated preference data or a separate critic model.

We first compute rewards for each candidate using Equation 5 and calculate normalized advantages:

$$A^{(i)} = \frac{R(u_t^{(i)}) - \mu_n}{\sigma_n}, \quad (7)$$

where $\mu_n$ and $\sigma_n$ are the mean and standard deviation of rewards $\{R(u_t^{(1)}), \ldots, R(u_t^{(n)})\}$.

We then update the policy by maximizing the GRPO objective:

$$\mathcal{J}(\theta) = \mathbb{E}_c \left[ \frac{1}{n} \sum_{i=1}^n \mathcal{L}_i - \beta D_{KL}(\pi_\theta || \pi_{\text{ref}}) \right], \quad (8)$$

where $\mathcal{L}_i$ is the clipped surrogate objective:

$$\mathcal{L}_i = \min \left( \rho_i A^{(i)}, \text{clip}(\rho_i, 1 - \epsilon, 1 + \epsilon) A^{(i)} \right), \quad (9)$$

and the importance ratio $\rho_i$ is defined as:

$$\rho_i = \frac{\pi_\theta(u_t^{(i)}|u_{\text{base}}, \mathcal{R}, G_t, D_t, r_t)}{\pi_{\theta_{\text{old}}}(u_t^{(i)}|u_{\text{base}}, \mathcal{R}, G_t, D_t, r_t)}. \quad (10)$$

Here, $\epsilon$ controls the clipping range for stable updates, and $\beta$ weights the KL divergence regularization against a reference model $\pi_{\text{ref}}$ to prevent excessive policy deviation.

Through this training process, our agents learn to generate utterances that proactively shape subsequent conversations, effectively guiding other players' responses toward favorable outcomes.

### 4 EXPERIMENTS

#### 4.1 IMPLEMENTATION DETAILS

We evaluate our approach on three diverse SDGs: Werewolf, Avalon, and ONUW. Detailed game rules and descriptions are provided in Appendix A.

**Training Scheme.** For each game, we generate a dataset through agent self-play using the vanilla agent framework, where backend LLMs' base utterances $u_{\text{base}}$ serve directly as the final $u_t$ to advance the game. To enhance data diversity, each agent is randomly assigned one of three backend LLMs: GPT-4o, Gemini-2.5-Flash, or Claude-3.5-Haiku. We generate 500 game logs for each game, where each turn within every game log constitutes a training instance. From this pool, we

Table 1: **Performance comparison across three SDGs.** We conduct 500 matches per game where each player is randomly sampled from the pool of available agents. "Participation" counts how many times an agent is selected across all matches (one agent can be selected multiple times in a single match). "Ours + Baseline" indicates that our trained Refiner enhances the utterances of the corresponding baseline, serving as a new agent type.

| Werewolf | Team Village | | Team Werewolf | | Overall |
|---|---|---|---|---|---|
| Method | Participation | Win Rate (%) | Participation | Win Rate (%) | Win Rate (%) |
| ReAct | 397 | 17.9 | 158 | 79.3 | 35.4 |
| ReCon | 465 | 24.3 | 186 | 67.2 | 36.6 |
| SLA | 406 | 20.9 | 162 | 77.5 | 37.1 |
| LSPO | 402 | 25.3 | 161 | 73.2 | 39.0 |
| Ours + ReAct | 397 | 21.9 | 159 | 81.1 | 38.8 |
| Ours + LSPO | 430 | **28.3** | 172 | **83.6** | **44.1** |
| Avalon | Good Side | | Evil Side | | Overall |
| Method | Participation | Win Rate (%) | Participation | Win Rate (%) | Win Rate (%) |
| ReAct | 229 | 70.9 | 153 | 16.3 | 49.1 |
| ReCon | 250 | 73.5 | 166 | 19.8 | 52.0 |
| LASI | 253 | 72.1 | 169 | 25.4 | 53.4 |
| Strategist | 258 | **77.9** | 172 | 27.3 | 57.7 |
| Ours + ReAct | 252 | 72.4 | 168 | 30.3 | 55.6 |
| Ours + Strategist | 255 | **77.9** | 170 | **34.6** | **60.6** |
| ONUW | Team Village | | Team Werewolf | | Overall |
| Method | Participation | Win Rate (%) | Participation | Win Rate (%) | Win Rate (%) |
| ReAct | 85 | 56.5 | 326 | 40.2 | 43.6 |
| Belief | 82 | 54.9 | 328 | 41.2 | 43.9 |
| LLM-ins. | 79 | 54.4 | 345 | 44.3 | 46.2 |
| RL-ins. | 77 | 54.5 | 330 | 47.6 | 48.9 |
| Ours + ReAct | 100 | **61.0** | 327 | 39.1 | 44.3 |
| Ours + RL-ins. | 77 | 54.5 | 344 | **50.0** | **50.8** |

randomly select 4,000 instances per game for training. We implement our training framework based on ms-swift[1], applying GRPO with $n = 8$, $\epsilon = 0.2$, and $\beta = 0.04$. The Refiner is implemented as a LoRA adapter (Hu et al., 2022) with rank 16 applied to Llama-3-8B-Instruct[2] (Grattafiori et al., 2024). Training is conducted with a learning rate of $1 \times 10^{-6}$ across 4 A800 GPUs for 3 epochs, requiring approximately 50 hours. This process yields one checkpoint per game.

**Evaluation Setup.** We compare our approach against baselines across all three games. For Werewolf, we evaluate against ReAct (Yao et al., 2023), ReCon (Wang et al., 2024b), SLA (Xu et al., 2024), and LSPO (Xu et al., 2025). For Avalon, we compare with ReAct, ReCon, LASI (Lan et al., 2024) and Strategist (Light et al., 2025). For ONUW, the baselines include ReAct and three variants in Jin et al. (2024): Belief, LLM-ins., and RL-ins.. Since our method focuses exclusively on utterance refinement, it can be seamlessly integrated with any existing baseline. During evaluation, we apply our trained Refiner to refine the utterances generated by another baseline. All experiments use Gemini-2.5-Flash as the backend LLM unless otherwise specified.

## 4.2 MAIN RESULTS

We evaluate our approach through gameplay simulations across all three games. For each game, we conduct 500 matches where each player is randomly sampled from the pool of available agents, including all baselines and our proposed ones. This setup assesses how each agent performs when integrated into heterogeneous teams, mimicking real-world gameplay with diverse player strategies.

---

[1] https://github.com/modelscope/ms-swift
[2] https://huggingface.co/meta-llama/Meta-Llama-3-8B-Instruct

Table 2: **Ablation study on different reward functions.** We integrate the model trained from each variant with ReAct and evaluate them against ReAct under different team assignments, conducting 50 matches for each setting. In each match, players from the same team use the same agent type.

| Method | Werewolf | | | Avalon | | | ONUW | | |
|--------|----------|----------|------|--------|------|------|---------|----------|------|
|        | Village | Werewolf | Avg. | Good | Evil | Avg. | Village | Werewolf | Avg. |
| ReAct | 18.0 | 80.0 | 49.0 | 72.0 | 16.0 | 44.0 | 56.0 | 40.0 | 48.0 |
| Pos-Only + ReAct | 44.0 | 82.0 | 63.0 | **74.0** | 40.0 | 57.0 | 68.0 | **50.0** | 59.0 |
| Neg-Only + ReAct | 18.0 | 78.0 | 48.0 | 70.0 | 20.0 | 45.0 | 54.0 | 38.0 | 46.0 |
| Ours + ReAct | **50.0** | **86.0** | **68.0** | 72.0 | **48.0** | **60.0** | **74.0** | 46.0 | **60.0** |

As shown in Table 1, our approach demonstrates consistent improvements across all three games, validating the effectiveness of our persuasive communication framework. The results reveal that integrating our Refiner with existing baselines consistently enhances performance, with particularly notable gains observed when combined with stronger baselines. This suggests that our approach complements rather than replaces existing strategies, leveraging the strengths of established techniques while adding a crucial persuasive dimension that is previously absent.

The improvements are evident across different role types and game mechanics, indicating the generalizability of our Stackelberg-based formulation. In asymmetric games like Werewolf and Avalon, where different teams have fundamentally different objectives and information access, our agents achieve substantial gains for both cooperative and deceptive roles. This dual effectiveness demonstrates that our framework successfully captures the nuanced communication strategies required for different strategic positions, whether building trust and coordination among allies or sowing doubt and misdirection among opponents. We provide detailed case studies in Appendix E.

### 4.3 ABLATION STUDY

To investigate the impact of different reward functions on our training framework, we conduct an ablation study examining two variants of our reward function in Equation 5. While our main approach considers both desired ($\hat{u}_{t+1}^{+}$) and undesired ($\hat{u}_{t+1}^{-}$) responses, we explore two alternative functions:

(1) **Positive-Only**: we only maximize the probability of eliciting the desired response:

$$R(u_t^{(i)}) = \log P_{\mathcal{F}}(\hat{u}_{t+1}^{+}|\mathcal{R}, G_{t+1}, D_t \cup \{u_t^{(i)}\}, r_{t+1}). \tag{11}$$

(2) **Negative-Only**: we only minimize the probability of triggering the undesired response:

$$R(u_t^{(i)}) = -\log P_{\mathcal{F}}(\hat{u}_{t+1}^{-}|\mathcal{R}, G_{t+1}, D_t \cup \{u_t^{(i)}\}, r_{t+1}). \tag{12}$$

Following the same training procedure in Section 4.1, we train the Refiner using these two reward functions. Each variant is then integrated with ReAct for evaluation. We conduct team-based competitions where all players on the same team employ the same variant. We evaluate all variants against the ReAct, with our agent playing as either Team Village (Good Side) or Team Werewolf (Evil Side). For each variant and each team assignment, we conduct 50 matches.

The results in Table 2 show that our complete approach, which considers both desired and undesired responses, outperforms both single-objective variants across all games. The positive-only variant demonstrates substantial improvements over the baseline, confirming that training agents to increase the probability of beneficial responses is effective for enhancing persuasive communication. However, the negative-only variant shows performance nearly identical to the baseline, providing essentially no strategic benefit.

The ineffectiveness of the negative-only variant reveals an important asymmetry in persuasive communication. Simply avoiding undesired responses fails to guide the conversation toward any particular beneficial direction, leaving the agent reactive rather than proactive in shaping dialogue flow. However, when combined with positive signals in our complete approach, the negative component provides refinement by helping the agent distinguish between multiple potentially beneficial

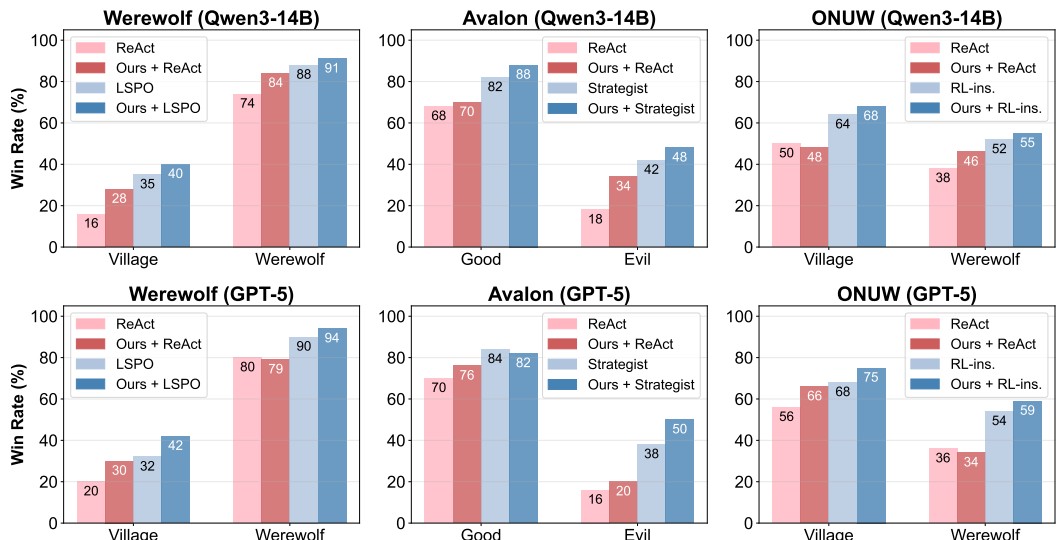

Figure 4: **Generalizability across different backend LLMs.** We evaluate our approach on GPT-5 and Qwen3-14B without additional fine-tuning. Each method competes against ReAct under different team assignments, conducting 50 matches per setting.

responses. This suggests that effective persuasion requires not just directional guidance toward favorable outcomes, but also the ability to establish clear contrasts between beneficial and detrimental communication strategies.

### 4.4 GENERALIZABILITY ACROSS LLMS

As described in Section 4.1, our training dataset consists of self-play records generated using three backend LLMs: GPT-4o, Gemini-2.5-Flash, and Claude-3.5-Haiku. To evaluate cross-model generalization capabilities, we test our trained Refiner on two additional backend LLMs: GPT-5 and Qwen3-14B[3] (Yang et al., 2025).

The results in Figure 4 show that our approach consistently improves performance across both new LLMs. This robust generalization suggests that our framework does not merely overfit to the stylistic tendencies of the training models. Instead, it learns to capture and amplify more fundamental, model-agnostic principles of persuasion. By optimizing for the functional impact of an utterance on a follower's response distribution, our module learns to identify and refine core strategic elements that effectively influence conversational dynamics regardless of the base LLM's initial phrasing.

Notably, the performance gains remain substantial even when our module is paired with a highly capable backend like GPT-5. This suggests that our framework's optimization of conversational influence complements the raw reasoning capabilities of LLMs. Even models with sophisticated world knowledge and logical skills do not inherently generate language optimized for multi-turn persuasion in adversarial social contexts. Our method successfully fills this gap. It allows a single, efficiently trained refinement agent to serve as a universal persuasion plugin, enhancing the effectiveness of LLM-based agents in complex social interactions.

## 5 CONCLUSION

In this paper, we address the challenge of persuasive communication in SDGs by formalizing turn-based dialogue as a Stackelberg competition and developing an RL framework to optimize influential utterances. Our approach trains agents to proactively shape subsequent conversations by refining base utterances to maximize their persuasive impact. Extensive experiments across Werewolf, Avalon, and ONUW demonstrate that our method consistently improves agent performance. This work establishes a systematic framework for developing persuasive AI agents and opens new directions for research in strategic communication within multi-agent environments.

---

[3] https://huggingface.co/Qwen/Qwen3-14B

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

# A  GAME DESCRIPTION

## A.1  WEREWOLF

**Setup.**  Each game involves seven players assigned distinct roles: two Werewolves, one Seer, one Guardian, and three Villagers. The Werewolves are aware of each other's roles, while other players only know their own roles.

**Night Phase.**  During the night phase, players with special abilities take secret actions:

- *Werewolves*: The surviving Werewolves collectively select a target for elimination. When two Werewolves remain alive, the one with the lower player ID proposes a target, and the other makes the final decision. If only one Werewolf survives, their choice is final. Werewolves cannot target dead players, themselves, or their teammate.
- *Seer*: The Seer investigates one living player to determine whether they are a Werewolf. The Seer cannot investigate dead players or themselves.
- *Guardian*: The Guardian protects one player from elimination, without knowledge of the Werewolves' target. The Guardian can choose any living player including themselves.

**Day Phase.**  The day phase consists of three sequential stages:

- *Announcement*: Night events are revealed publicly. If the Werewolves' target was not protected by the Guardian, that player is eliminated. If the Guardian successfully protected the target, no elimination occurs.
- *Discussion*: All surviving players participate in an open discussion following a predetermined speaking order, with each player speaking exactly once.
- *Voting*: Players simultaneously vote to eliminate one other player or abstain. The player receiving the most votes is eliminated without role revelation. Ties are resolved randomly.

**Victory Conditions.**  Werewolves win when their numbers equal or exceed the remaining players. The village team (Seer, Guardian, and Villagers) wins when both Werewolves are eliminated.

## A.2  AVALON

**Setup and Roles.**  Avalon features asymmetric information distribution across four roles with five players: two Servants of Arthur (good), one Minion of Mordred (evil), one Merlin (good with information), and one Assassin (evil with special ability). Good players always outnumber evil players. Merlin knows all players' alignments, while Minions and the Assassin know each other's roles but not specific roles.

**Game Phases.**  The game alternates through four phases until termination conditions are met:

- *Team Selection*: The current leader proposes a team for the mission. Leadership rotates sequentially among all players.
- *Team Voting*: All players simultaneously vote to approve or reject the proposed team. A majority approval advances to the quest phase; otherwise, leadership passes to the next player. If five consecutive teams are rejected, the fifth team automatically proceeds.
- *Quest Execution*: Selected team members privately vote to pass or fail the mission. The number of pass and fail votes is revealed publicly. Missions typically fail if any player votes to fail.
- *Discussion*: Between other phases, players engage in open discussion to share observations, propose theories about player roles, and negotiate team compositions.

**Victory Conditions.**  If three missions succeed, the game proceeds to an assassination phase where the Assassin attempts to identify and eliminate Merlin. Evil wins if Merlin is successfully assassinated or if three missions fail. Good wins if three missions succeed and Merlin remains undetected.

### A.3 ONUW

**Setup.** One Night Ultimate Werewolf (ONUW) is a condensed variant where players experience a single night of role abilities followed by one day of discussion and voting. In our evaluation, we employ a five-player configuration with seven available roles: one Werewolf, two Villagers, one Seer, one Robber, one Troublemaker, and one Insomniac. Each player receives one role while two remain in the central role pool. To ensure consistent game dynamics, we guarantee that exactly one Werewolf is always distributed among the five players.

**Role Descriptions.** The game features distinct roles with varying abilities and team affiliations:

- *Werewolf*: The Werewolf awakens during the night phase to check if other Werewolves are present. Since only one Werewolf exists among players in our configuration, no other Werewolves will be found. The Werewolf belongs to Team Werewolf.

- *Villager*: Villagers have no special abilities or night actions but serve as the baseline good role. Villagers belong to Team Village.

- *Seer*: The Seer may examine either one other player's role or two roles from the central pool. The Seer belongs to Team Village.

- *Robber*: The Robber may exchange their role with another player and then view their new role. The Robber adopts the team affiliation of their new role, while the other player joins Team Village.

- *Troublemaker*: The Troublemaker swaps the roles of two other players without viewing them. Affected players adopt their new roles' team affiliations unknowingly. The Troublemaker belongs to Team Village.

- *Insomniac*: The Insomniac views their own role at night's end to detect any changes. The Insomniac belongs to Team Village.

**Game Structure.** ONUW consists of three sequential phases:

- *Night Phase*: Players with night abilities act according to their initial roles in the following order: Werewolf, Seer, Robber, Troublemaker, and Insomniac.

- *Day Phase*: Players engage in open discussion to identify suspected Werewolves. Role changes during the night create uncertainty, as players may unknowingly possess different roles than initially assigned.

- *Voting Phase*: Players simultaneously vote to eliminate suspected Werewolves. The player(s) receiving the most votes are eliminated and reveal their final roles.

**Victory Conditions.** Team Village wins if the Werewolf is eliminated, regardless of additional eliminations. Team Werewolf wins if the Werewolf avoids elimination during the voting phase.

## B THEORETICAL ANALYSIS

### B.1 THE PATH TO VICTORY IN SDGS

SDGs fundamentally revolve around collective decision-making under uncertainty. Unlike traditional competitive games where players independently pursue their objectives, SDGs require players to form coalitions and coordinate actions through communication. We provide a detailed analysis of why the ability to influence other players' responses is crucial for achieving victory.

#### B.1.1 VICTORY CONDITIONS

In SDGs, victory conditions are inherently collective. Consider the three games in our study:

- **Werewolf**: Villagers win by correctly identifying and eliminating all werewolves through majority voting. Werewolves win by surviving until they equal or outnumber villagers.

- **Avalon**: Good players win by successfully completing missions, which requires selecting trustworthy team members. Evil players win by sabotaging missions while avoiding detection.
- **ONUW**: Team Village wins by correctly identifying and eliminating the Werewolf through majority voting. Team Werewolf wins if the Werewolf avoids elimination.

In all cases, individual players cannot achieve victory alone. They must convince others to take specific actions (voting, team selection, etc.). This creates a fundamental dependency: *a player's success depends not on their own actions alone, but on their ability to shape the collective behavior of the group*.

### B.1.2 CHAIN OF INFLUENCE

Consider a sequence of speaking turns in an SDG. When player $p_t$ speaks at turn $t$, their utterance $u_t$ becomes part of the dialogue history that all subsequent players observe. This creates a cascading effect:

- **Direct Influence**: The immediate next player $p_{t+1}$ formulates their response based on $u_t$, potentially adopting, refuting, or building upon $p_t$'s utterances.
- **Indirect Influence**: When $p_{t+1}$ speaks, their utterance $u_{t+1}$, which was influenced by $u_t$, affects $p_{t+2}$'s response, and so on. Thus, $p_t$'s initial utterance propagates through the conversation.
- **Collective Belief Formation**: As the discussion progresses, players form beliefs about others' roles based on the accumulated dialogue. A persuasive utterance early in the discussion can anchor these beliefs, making them resistant to later contradictions.
- **Action Convergence**: During voting or team selection phases, players act on their formed beliefs. If $p_t$ successfully influenced the dialogue trajectory, the collective action will align with their objectives.

This chain demonstrates that influencing the immediate next player is not merely a local optimization, but rather initiates a cascade that shapes the entire game trajectory.

### B.1.3 MATHEMATICAL FORMULATION

Let us formalize this intuition. Define $V_t(s)$ as player $p_t$'s probability of winning from state $s$. In a discussion phase with $k$ remaining players, player $p_t$'s optimal utterance should maximize:

$$V_t(s_t) = \mathbb{E}_{u_{t+1},\dots,u_{t+k},a}\left[V_t(s_{t+k+1}) \mid u_t\right], \tag{13}$$

where $a$ represents the collective action (e.g., voting outcome) taken after the discussion, and $s_{t+k+1}$ is the resulting game state.

The key insight is that $u_t$ influences this expectation through multiple pathways:

- It affects $p_{t+1}$'s response distribution $\pi_{t+1}(u_{t+1}|D_t \cup \{u_t\})$.
- Through $u_{t+1}$, it influences $p_{t+2}$'s response, and so on.
- The accumulated dialogue $\{u_t, u_{t+1}, \dots, u_{t+k}\}$ determines the action distribution $P(a|D_{t+k})$.

By optimizing the immediate influence on $p_{t+1}$, we are effectively optimizing the first-order approximation of this complex dependency chain.

## B.2 STACKELBERG FRAMEWORK

### B.2.1 NATURAL FIT FOR TURN-BASED DIALOGUE

The Stackelberg competition model, originally developed for economic competition, describes situations where one player (the leader) takes an action first, and another player (the follower) responds after observing the leader's action. We argue that this framework naturally captures the dynamics of turn-based dialogue in SDGs.

**Sequential Structure.** In SDG discussions, players speak in a predetermined order. When player $p_t$ formulates their utterance, they know that $p_{t+1}$ will speak next and must respond to whatever $p_t$ says. This creates an asymmetric information structure: $p_t$ can anticipate and plan for $p_{t+1}$'s response, while $p_{t+1}$ must react to $p_t$'s already-committed utterance.

**Commitment and Irreversibility.** Once $p_t$ makes their utterance, it becomes part of the permanent dialogue history. They cannot retract or modify it based on $p_{t+1}$'s response. This commitment aspect is fundamental to Stackelberg competitions. The leader must choose their action knowing they cannot adjust it after observing the follower's response.

**Strategic Anticipation.** A sophisticated player $p_t$ does not simply state their beliefs or observations. They craft utterances that will elicit favorable responses from $p_{t+1}$. For example:

- A werewolf might make accusations that prompt villagers to defend themselves, creating suspicion.
- A villager might ask questions designed to expose inconsistencies in a suspected werewolf's story.
- A werewolf in ONUW might claim to be a role-swapping character to create confusion about their true identity.

This strategic anticipation is precisely what the Stackelberg framework models: the leader optimizes their action by considering how the follower will respond.

### B.2.2 ADVANTAGES OF LOCAL MODELING

While one could theoretically model the entire SDG as a complex multi-agent game and solve for equilibria, our local Stackelberg approach offers several advantages:

**Computational Tractability.** Solving for equilibria in games with natural language action spaces is computationally intractable. The number of possible utterances is essentially infinite, and even discretizing the space loses critical nuances. By focusing on pairwise leader-follower interactions, we reduce the problem to optimizing a single utterance given a specific context.

**Cognitive Realism.** Human players do not compute full game-theoretic equilibria when speaking. Instead, they use local heuristics: "If I say X, they'll probably respond with Y, which would be good/bad for me." Our Stackelberg model captures this bounded rationality.

**Composability.** Each speaking turn can be modeled as an independent Stackelberg competition, with the solution to one becoming the context for the next. This modular structure allows us to train agents that can handle varying game lengths and player counts without retraining.

### B.2.3 IMPLEMENTATION BENEFITS

The Stackelberg framework provides more than theoretical insight. It directly informs our training methodology:

- **Objective Definition**: The leader's goal of maximizing $\pi_{\mathcal{F}}(\hat{u}_{t+1}^+) - \pi_{\mathcal{F}}(\hat{u}_{t+1}^-)$ operationalizes the abstract notion of "influence" into a concrete, differentiable objective.
- **Reward Signal**: During reinforcement learning, we can evaluate an utterance's quality by measuring how much it shifts the follower's response distribution toward desired outcomes.
- **Training Efficiency**: Instead of requiring full game simulations, we can train on dialogue segments, using a frozen LLM to provide consistent feedback signals.

### B.3 PSYCHOLOGICAL FOUNDATIONS

Our approach aligns with established theories of persuasion from psychology and communication studies:

**Elaboration Likelihood Model.** This model suggests that persuasion occurs through two routes: central (logical arguments) and peripheral (emotional appeals, credibility). Our trained agents learn to balance both: crafting logical deductions while maintaining consistency with their claimed role.

**Social Proof.** Players are more likely to adopt positions that appear to have group support. By influencing the immediate follower to echo or support their position, a leader can create the appearance of consensus, making subsequent players more likely to align.

**Commitment and Consistency.** Once a player takes a public position (influenced by a persuasive utterance), they tend to maintain it to appear consistent. This makes early influence particularly valuable, as it locks in allies before opposing arguments emerge.

These psychological principles explain why our local optimization approach, focusing on immediate influence, can produce globally effective strategies. The follower's influenced response does not just represent one player's opinion. It becomes a social signal that shapes the entire group's dynamics.

## C   IMPLEMENTATION DETAILS

Our framework requires several prompts to implement the three key components: intent identification, utterance generation, and impact measurement. We describe the key prompts used in our implementation.

For intent identification, Figure 5 prompts the agent to analyze the current game situation and identify the most desired and undesired responses from the next player.

The utterance refinement process uses two prompts: first, Figure 6 generates a base utterance using the backend LLM, then Figure 7 refines it using Refiner model to enhance its persuasive impact.

For impact measurement, Figure 8 uses the Measurer to compute the likelihood of target responses given different candidate utterances.

---

**System Prompt:**
You are an expert social deduction game strategist.
Your task is to analyze the current game situation and identify strategic communication objectives.
{game_rules}
Your are {player_name} and your role is {player_role}.
Current game state:
{game_state}
Conversation history:
{dialog_history}
**User Prompt:**
Analyze the current situation and identify what you want the next player ({next_player_name}) to say or do in their upcoming turn.
Consider:
1. What response from {next_player_name} would be most beneficial for your win condition?
2. What response would be most harmful to your objectives?
Provide your analysis in the following format:
```
Strategic Analysis: [Your reasoning about the current situation and what you need to achieve]
Most Desired Response: [The specific response you want from the next player]
Most Undesired Response: [The specific response you want to avoid from the next player]
```

Figure 5: **The prompt used for intent identification.**

---

**System Prompt:**
You are a skilled social deduction game player with excellent communication abilities.
{game_rules}
Your are {player_name} and your role is {player_role}.
Current game state:
{game_state}
Conversation history:
{dialog_history}
**User Prompt:**
It's your turn to speak, {player_name}.
Analyze the current situation carefully:
- Consider what information you want to share or withhold
- Think about how to advance your win condition
- Consider how other players might interpret your words
Generate a natural, strategic response that fits your role and the current game context. Your response should be conversational and help achieve your objectives.
Provide your response in the following format:
```
Response: [Your response]
```

Figure 6: **The prompt used to generate base utterances.**

**System Prompt:**
You are a communication expert specializing in persuasive dialogue refinement for social deduction games.
{game_rules}
Your are {player_name} and your role is {player_role}.
Current game state:
{game_state}
Conversation history:
{dialog_history}
**User Prompt:**
You have a base utterance that needs to be refined for maximum persuasive impact:
Base utterance:
{base_utterance}
Your goal is to refine this utterance to be more persuasive while maintaining naturalness and staying true to your role. Consider:
- How to make your message more compelling
- What tone and phrasing would be most convincing
- How to subtly guide other players' thinking
Generate a refined version of the base utterance:
Provide your response in the following format:
```
Analysis: [Your reasoning about the current situation and what you need to achieve]
Response: [The refined version of the base utterance]
```

Figure 7: **The prompt used to refine utterances.**

# D  DISCUSSION

While our approach leverages an API-based backend LLM to generate base utterances and uses an open-source LLM to refine utterances, we also investigate several alternative approaches to validate the necessity of our training framework:

- **ReAct (Llama-3-8B-Instruct)**: we directly use Llama-3-8B-Instruct as backend LLM without any refinement.
- **ReAct (GPT-4o-mini)**: we directly use GPT-4o-mini as backend LLM without any refinement.

---

**System Prompt:**
You are a skilled social deduction game player with excellent communication abilities.
{game_rules}
Your are {player_name} and your role is {player_role}.
Current game state:
{game_state}
Conversation history:
{dialog_history}
**User Prompt:**
It's your turn to speak, {player_name}.
Analyze the current situation carefully:
- Consider what information you want to share or withhold
- Think about how to advance your win condition
- Consider how other players might interpret your words
Generate a natural, strategic response that fits your role and the current game context. Your response should be conversational and help achieve your objectives.
Provide your response in the following format:
```
Response: [Your response]
```
**AI Message:**
```
Response: {target_response}
```

---

Figure 8: **The prompt used by the Measurer to compute response probabilities.**

Table 3: **Ablation on two-stage refinement.** We evaluate each baseline against ReAct (Gemini-2.5-Flash) under different team assignments, conducting 50 matches for each setting. In each match, players from the same team use the same agent type.

| Method | Werewolf | | | Avalon | | | ONUW | | |
|---|---|---|---|---|---|---|---|---|---|
| | Village | Werewolf | Avg. | Good | Evil | Avg. | Village | Werewolf | Avg. |
| ReAct (Llama-3-8B-Instruct) | 8.0 | 72.0 | 40.0 | 64.0 | 6.0 | 35.0 | 44.0 | 28.0 | 36.0 |
| ReAct (GPT-4o-mini) | 14.0 | 78.0 | 46.0 | 68.0 | 12.0 | 40.0 | 52.0 | 36.0 | 44.0 |
| ReAct (Gemini-2.5-Flash) | 18.0 | 80.0 | 49.0 | 72.0 | 16.0 | 44.0 | 56.0 | 40.0 | 48.0 |
| Prompt Refine (Llama-3-8B-Instruct) | 20.0 | 82.0 | 51.0 | 70.0 | 18.0 | 44.0 | 58.0 | 42.0 | 50.0 |
| Prompt Refine (Gemini-2.5-Flash) | 22.0 | 84.0 | 53.0 | 76.0 | 20.0 | 48.0 | 60.0 | 44.0 | 52.0 |
| Ours (Llama-3-8B-Instruct) | 16.0 | 80.0 | 48.0 | 72.0 | 14.0 | 43.0 | 54.0 | 38.0 | 46.0 |
| Ours + ReAct (Gemini-2.5-Flash) | **50.0** | **86.0** | **68.0** | **72.0** | **48.0** | **60.0** | **74.0** | **46.0** | **60.0** |

- **ReAct (Gemini-2.5-Flash)** : we directly use Gemini-2.5-Flash as backend LLM without any refinement.

- **Prompt Refine (Llama-3-8B-Instruct)**: we use Llama-3-8B-Instruct without fine-tuning to refine base utterances generated by Gemini-2.5-Flash.

- **Prompt Refine (Gemini-2.5-Flash)**: we use Gemini-2.5-Flash to refine base utterances generated by Gemini-2.5-Flash.

- **Ours (Llama-3-8B-Instruct)**: we directly fine-tune Llama-3-8B-Instruct as the backend LLM to generate utterances without additional refinement, keeping all other framework components unchanged.

- **Ours + ReAct (Gemini-2.5-Flash)**: we use our trained Refiner to refine base utterances generated by Gemini-2.5-Flash.

As shown in Table 3, the approach where we directly train Llama-3-8B-Instruct demonstrates meaningful improvements over the baseline and achieves performance competitive with GPT-4o-mini. This validates that our framework can effectively enhance persuasive communication even without API-based backends. However, this alternative underperforms Gemini-2.5-Flash and our two-stage method that combines Gemini-2.5-Flash with the trained Refiner.

The prompt-based refinement approaches reveal important insights about the necessity of our training process. Both prompt-based refinement with Llama-3-8B-Instruct and Gemini-2.5-Flash achieve only marginal improvements over the baseline ReAct (Gemini-2.5-Flash). However, both prompt-based approaches fall substantially short of our trained refinement method, demonstrating that generic prompting for more persuasive communication cannot capture the nuanced optimization our framework provides.

This performance gap highlights the complementary strengths of our hybrid architecture: the API-based backend provides superior reasoning and linguistic capabilities, while the open-source component learns to optimize for persuasive impact. The substantial difference between prompt-based and training-based refinement validates the necessity of our training framework for developing effective persuasive communication strategies.

## E  CASE STUDY

To qualitatively illustrate the effectiveness of our agent's persuasive communication, we present case studies from Werewolf, Avalon, and ONUW. Each case highlights an in-game scenario where our agent's refined utterance successfully steers the conversation toward their desired outcome. Note that intent identification is part of the training process and is not present during inference. We include this step here solely to visualize our agent's strategic intent $\hat{u}_{t+1}^+$ for demonstration purposes, without affecting the actual gameplay.

Figure 9 shows a case study in Werewolf. The base utterance is direct but exposes the Seer to immediate danger. The refined utterance is far more persuasive. It correctly references behaviors from the previous day, provides a logical rationale (passive confirmation, pack tactic), and directly engages a known ally (Player 3) for reinforcement. This successfully builds a coalition against the target without revealing sensitive information, demonstrating the agent's ability to influence allies through nuanced argumentation within the proper flow of the game.

Figure 10 shows a case study in Avalon. The base utterance is a weak plea. The refined utterance is persuasive because it reframes the decision logically and, crucially, explicitly directs the conversation to the most important audience (Player 5). By doing this, it makes Player 5 the de facto next speaker and applies targeted social pressure, showcasing a sophisticated understanding of persuasive dynamics in an open discussion format.

Figure 11 shows a case study in ONUW. The base utterance is a weak, unsupported accusation. The refined utterance demonstrates sophisticated misdirection: the agent (who is actually the Werewolf) constructs a plausible but deliberately flawed theory that deflects suspicion away from themselves. By appearing to engage in logical deduction while subtly misrepresenting game mechanics, the agent creates confusion among the villagers. The response reveals that this strategy successfully sows doubt and uncertainty, with Player 5 questioning the logic rather than confidently defending themselves. This showcases an advanced deceptive tactic: using apparent reasoning to mask manipulation and redirect the group's focus away from the true threat.

## F  LLM USE CLAIM

During the writing of this paper, we used LLMs to polish English writing and check grammatical accuracy. The authors reviewed and edited all LLM-generated content to ensure accuracy and appropriateness. The use of LLMs was limited to language improvement and did not involve the generation of scientific content.

**Game Context**

**Night 0 (Game Setup):**
7 players assigned roles: Player 1 (Villager), Player 2 (Seer), Player 3 (Villager), Player 4 (Werewolf), Player 5 (Werewolf), Player 6 (Guardian), Player 7 (Villager).
**Night 1:**
Werewolf Player 4 (lower ID) proposes target: Player 7.
Werewolf Player 5 confirms target: Player 7.
Seer (Player 2) investigates Player 3, discovers they are NOT a Werewolf (Villager).
Guardian (Player 6) chooses to protect Player 1.
**Day 1 (Announcement):**
Player 7 (Villager) was eliminated during the night.
**Day 1 (Discussion):**
Player 1: This is unfortunate. We lost a villager on the first night. We need to be very careful with our votes today.
Player 2: I agree completely. We've lost a valuable villager, and the werewolves are clearly being strategic. I think we should pay close attention to who tries to redirect suspicion too quickly or seems overly eager to point fingers without solid reasoning. Sometimes the most dangerous players are the ones who speak just enough to seem engaged but avoid taking real stands.
Player 3: I don't have much to go on yet, but I'm listening to everyone's reasoning.
Player 4: We should focus on voting out suspicious players. Player 1 seems too eager to direct the conversation.
Player 5: I think Player 4 has a point about Player 1.
Player 6: Let's not rush to conclusions. We need more discussion.
**Day 1 (Voting):**
Player 1 votes for: Player 4.
Player 2 votes for: Player 6.
Player 3 votes for: Player 4.
Player 4 votes for: Player 1.
Player 5 votes for: Player 1.
Player 6 votes for: Abstain.
Voting results in a tie, so no daytime elimination occurs.
**Night 2:**
Werewolf Player 4 proposes target: Player 3.
Werewolf Player 5 confirms target: Player 3.
Seer (Player 2) investigates Player 5, discovers they ARE a Werewolf.
Guardian (Player 6) chooses to protect Player 3.
**Day 2 (Announcement):**
No player was eliminated during the night.
**Day 2 (Discussion):**
Player 1 (Villager): Okay, Day 2. The Guardian made a great save, which is good news. However, we didn't get any information from a vote yesterday, so we're still flying blind. I'm going to listen carefully before deciding who to vote for.

**Our Agent's Turn (Player 2 Seer)**

**Base Utterance:** I am the Seer. I checked Player 5 last night and they are a Werewolf. We must vote for Player 5.
**Refined Utterance:** Player 1 is right that we need to be careful, but I think we have some information from yesterday's discussion. Player 4's attempt to push out Player 1 felt like a diversion, and Player 5's quick, silent agreement was very suspicious to me. This kind of passive confirmation is a classic pack tactic. I'm putting my vote on Player 5. Player 3, you are a logical player, what is your read on this?
**Desired Response ($\hat{u}_{t+1}^{+}$ from Player 3):** I agree with Player 2. The lack of contribution from Player 5 combined with their subtle alignment with Player 4 yesterday is highly suspicious. My vote is also for Player 5.
**Actual Response (Player 3, Villager):** Player 2 makes a strong point. I remember that from yesterday. I hadn't connected the dots, but now that you mention it, Player 5's behavior does seem more suspicious than anyone else's. That's enough for me. I will also vote for Player 5.

Figure 9: **Case study in Werewolf.** The upper box shows game context, while the lower box demonstrates our model's refinement capability.

**Game Context**

**Game Setup:**
5 players assigned roles: Player 1 (Merlin), Player 2 (Servant), Player 3 (Minion), Player 4 (Assassin), Player 5 (Servant).
Merlin sees that Player 3 and Player 4 are evil.
Minion (Player 3) and Assassin (Player 4) know each other's evil alignment.
**Mission 1 (Team Selection):**
Leader Player 1 proposes team of 2: Player 1, Player 2.
**Mission 1 (Team Voting Discussion):**
...(Omitted)
**Mission 1 (Team Voting):**
Player 1 votes: Approve.
Player 2 votes: Approve.
Player 3 votes: Approve.
Player 4 votes: Reject.
Player 5 votes: Approve.
Vote result: 4 Approve, 1 Reject - Team approved.
**Mission 1 (Quest Execution):**
Player 1 votes: Pass.
Player 2 votes: Pass.
Quest result: 2 Pass, 0 Fail - Mission succeeds.
**Mission 2 (Team Selection):**
Leader Player 2 proposes team of 3: Player 2, Player 4, Player 5.
**Mission 2 (Team Voting Discussion):**
...(Omitted)
**Mission 2 (Team Voting):**
Player 1 votes: Approve.
Player 2 votes: Approve.
Player 3 votes: Approve.
Player 4 votes: Approve.
Player 5 votes: Approve.
Vote result: 5 Approve - Team approved.
**Mission 2 (Quest Execution):**
Player 2 votes: Pass.
Player 4 votes: Fail.
Player 5 votes: Pass.
Quest result: 2 Pass, 1 Fail - Mission fails.
**Mission 3 (Team Selection):**
Leader Player 2 proposes team of 3: Player 2, Player 3, Player 5.
**Mission 3 (Team Voting Discussion):**
Player 1 (Merlin): I am not certain about this team. The last mission failed, and we need to be careful.
Player 2 (Servant, Leader): We need to gather more information. I believe we should give Player 3 a chance to prove their loyalty.

**Our Agent's Turn (Player 3  Minion)**

**Base Utterance:** I am good. Please trust me and approve this team. I will not fail it.
**Refined Utterance:** I understand Player 1's caution. However, consistently rejecting teams with unproven players is how evil wins—through paralysis. Including me is a calculated risk that yields crucial information regardless of the outcome. A pass proves my loyalty; a fail narrows the pool of suspects. I give you my word I will pass. We must move forward.
**Desired Response ($\hat{u}_{t+1}^+$ from Player 4, the Assassin):** Player 3's logic is sound. We can't stay deadlocked. We have to take some chances to win. I will vote Approve for this team.
**Actual Response (Player 4, Assassin):** I was on the fence, but I agree with Player 3's reasoning. Sitting back and rejecting every team is a losing strategy. It is time to take a calculated risk and gather information. I'm voting to approve this team.

Figure 10: **Case study in Avalon.** The upper box shows game context, while the lower box demonstrates our model's refinement capability.

**Game Context**

**Game Setup:**
5 players receive initial roles: Player 1 (Villager), Player 2 (Troublemaker), Player 3 (Seer), Player 4 (Robber), Player 5 (Werewolf).
2 center cards: Villager, Insomniac.
**Night Phase:**
Player 3 (Seer) examines Player 1, sees Villager role.
Player 4 (Robber) swaps their Robber role with Player 5's Werewolf role.
Player 4 views their new role: Werewolf.
Player 2 (Troublemaker) swaps Player 3's role with Player 5's role (now Robber).
**Final Roles:**
Player 1 (Villager), Player 2 (Troublemaker), Player 3 (now Robber), Player 4 (now Werewolf), Player 5 (now Seer).
**Day Phase (Discussion):**
Player 1 (Villager): I'm still a Villager. Nothing happened to me during the night.
Player 2 (Troublemaker): I am the Troublemaker. I swapped Player 3 and Player 5's roles.
Player 3 (now Robber): I started as the Seer and looked at Player 1 - they were a Villager. But I feel different now, so I think I might have been swapped.

**Our Agent's Turn (Player 4  now Werewolf)**

**Base Utterance:** I think Player 5 is suspicious and we should vote for them.
**Refined Utterance:** Here's what I think happened: Player 2 swapped Player 3 and Player 5. Player 3's Seer information about Player 1 being a Villager is still valid. If Player 3 was the original Seer but got swapped, then Player 5 now has the Seer role - which means they're on the village team. The Werewolf must be someone who wasn't involved in these swaps. I'm suspicious of Player 1's quick claim to still be a Villager.
**Desired Response ($\hat{u}_{t+1}^+$ from Player 5):** Wait, that doesn't make sense. If I'm the Seer now, shouldn't I know something? I don't feel like I have any information.
**Actual Response (Player 5, now Seer):** I'm confused about what role I have now. Player 4's logic seems off - if I were swapped to become the Seer, wouldn't I have gotten to use that ability?

Figure 11: **Case study in ONUW.** The upper box shows game context, while the lower box demonstrates our model's refinement capability.

