# OpenReview forum: "Leading the Follower: Learning Persuasive Agents in Social Deduction Games"
_ICLR.cc/2026/Conference — ICLR 2026 Conference Withdrawn Submission_

### Official Review · Reviewer_nFNw · 2025-10-30

**Soundness:** 3
**Presentation:** 2
**Contribution:** 3
**Rating:** 4
**Confidence:** 3

**Summary:**

This paper frames turn-based dialogue as a Stackelberg competition and develops a method that is able to classify intents and use this to create more complicated rewards. The paper presents novel techniques to measure the impact of the shift in rewards and perform extensive experiments highlighting the effectiveness of their technique on social deduction games. The intention modelling is a very exciting direction.

**Strengths:**

1. The formulation of the multi-turn multi-agent problem as a Stackelberg competition is novel and interesting
2. The algorithm provides clear performance improvements over all three environments
3. The intention detection and formulation of shifting the “listener” model towards intended utterances and beliefs is very exciting!
4. Using a “refiner” open source model on top of a closed source model is a nice idea because it allows us to use the underlying power of a large model, combined with a small adapter fine-tuning.
5. The paper performs an ablation study in positive and negative rewards and shows that combining both desired and undesired responses is useful

**Weaknesses:**

1. Missing some related work on using similar techniques (e.g. RL for LLMs) on similar domains. Previous papers have explored using RL for persuasion or deception in clinical conversations [1], negotiation [2], and social environments [3]. Previously, people have used iterative rejection sampling (a simple version of reinforcement learning) on environments like SOTOPIA, with SOTOPIA-pi [4].
2. For me, this paper needs more motivation about “why strategic deduction games” instead of other similar benchmarks, and why developing algorithms for this could be more generalizable to real-life or other multi-agent ideas

[1] Feng, Yichun, et al. "Doctoragent-rl: A multi-agent collaborative reinforcement learning system for multi-turn clinical dialogue." arXiv preprint arXiv:2505.19630 (2025)
[2] Abdulhai, Marwa, et al. "Lmrl gym: Benchmarks for multi-turn reinforcement learning with language models." arXiv preprint arXiv:2311.18232 (2023).
[3] Yu, Haofei, et al. "Sotopia-RL: Reward Design for Social Intelligence." arXiv preprint arXiv:2508.03905 (2025).
[4] Wang, Ruiyi, et al. "SOTOPIA-$\pi $: Interactive Learning of Socially Intelligent Language Agents." arXiv preprint arXiv:2403.08715 (2024).

**Questions:**

1. Is this approach only applicable to social deduction games, or could this be a more general method around deception and inference?
2. The method seems to require access to a “follower” model. This seems like it would be difficult to do with humans. Do the authors believe that this could be generalized to human players?

**Details Of Ethics Concerns:**

This paper involves deception and learning social deduction. This could have malicious uses if LLMs are used to actively persuade or manipulate people online. I think an Ethics statement from the authors addressing these concerns would be warranted.

---

### Official Review · Reviewer_VNYw · 2025-10-31

**Soundness:** 2
**Presentation:** 3
**Contribution:** 2
**Rating:** 4
**Confidence:** 3

**Summary:**

This paper studies persuasive language in social deduction games (specifically Werewolf, Avalon, and One Night Ultimate Werewolf), that is, strategic games that have a large component of communication and influencing other player's beliefs. It formalizes this problem from a Stackelberg game perspective where the leader communicates words that optimally influences a follower's response distribution in the game. Its core contribution lies in a GRPO-based RL fine-tuning pipeline based on the Stackelberg framework. They demonstrate its effectiveness by training on a dataset obtained with LLM agent baselines in self-play, and by showing that the finetuned agent performs much better in terms of win rate.

**Strengths:**

- clear to understand and well-written paper about a generally interesting topic to the ML community
- experimented with various baseline LLMs and agent architectures

**Weaknesses:**

- Social deduction games are an interesting test bed of language games, and specifically for deception, persuasive language, and communication more generally. Nonetheless I find it quite narrow to study persuasive language in social deduction games only. Your methods have potential implications for persuasive language in strategic games more generally, and I think that would have been the better scope for this project. For example, does your method work well in games of bargaining, negotiation, or debate? Of course, this is not a critique that the authors can easily resolve at this stage

**Questions:**

- Do I understand it correctly that you greatly simplify the question of how to measure an agent's persuasive impact in your RL pipeline? Namely, you let an LLM judge come up with what would be the most advantageous and the most disadvantageous sentences the leader could outputting right now, and then you measure the difference in likelihood that an agent might return the most advantageous output (and only exactly that one, verbatim) vs the most disadvantageous one (again, only exactly that one). If this understanding is correct, I find that it could potentially be a very misleading measure (including your ablations in Section 4.3 where you only include the likelihood of one of the two ouptuts). A further discussion here would be highly appreciated.

**Details Of Ethics Concerns:**

This is work on how to improve social persuasion for the benefit of oneself, tested in strategic games where deception is a crucial component to success. The potentially unethical applications are evident

---

### Official Review · Reviewer_mi81 · 2025-11-01

**Soundness:** 3
**Presentation:** 3
**Contribution:** 3
**Rating:** 6
**Confidence:** 5

**Summary:**

This paper aims to learn persuasive agents in social deduction games like Werewolf and Avalon. While previous work mainly focuses on logical deduction and strategies, this work considers a different dimension of convincing others to respond following the speaker's intention. To do this, the authors collect a self-play dataset and train a Refiner that maximizes the follower's probability to generate favorable responses. Experiment results on three social deduction games show the proposed method improves performance and generalizes with different backends.

**Strengths:**

1. Clear motivation and idea: for SDG agents, prior work mainly focuses on improving logic and strategy. This paper, instead, considers a complementary yet equally important dimension of how to make an agent’s language more persuasive given a fixed strategy. The idea is both novel and reasonable.
2. Solid experimental results: The paper conducts extensive experiments across three representative SDGs and multiple baselines, achieving good performance. The ablation study on the reward design and the analysis of generalization across LLMs are both solid and insightful.
3. Good clarity and presentation: The manuscript is well written and easy to follow. Figures and tables effectively illustrate both the proposed method and experimental outcomes.

**Weaknesses:**

1. Fixed follower during training: my major concern is that the follower in training is a fixed model, whereas in multi-agent environments, other agents are often diverse and may co-evolve. This means the response distribution of followers could vary significantly and shift during training. As a result, training with a single fixed follower may lead to a leader that performs poorly when facing unseen or stylistically different agents.
2. Increased model complexity: the method introduces an additional model, the Refiner, to enhance persuasiveness. Compared to directly training a single agent, this adds extra model complexity and computational cost.
3. Limited reproducibility: the dataset and code have not been released, raising minor concerns about reproducibility.

**Questions:**

1. Regarding Weakness 1, could the authors analyze how a fixed follower limits adaptability? For example, comparing the response distributions of two stylistically distinct followers to the same refined utterance might help illustrate the issue. Would generating a diverse follower population mitigate this problem?
2. In Table 1, only the average win rate across different agents is reported. Could the authors also provide per-agent win rates? This would clarify whether the improvements are uniformly distributed or concentrated on certain follower types. Since ReAct is very similar to the training follower, the observed improvement might mainly stem from better alignment with that specific agent.
3. Could the proposed method be combined with existing methods that focus on optimizing logic and strategies to jointly train a single model that integrates both backend reasoning and linguistic refinement?

---

### Official Review · Reviewer_QGQ5 · 2025-11-12

**Soundness:** 2
**Presentation:** 2
**Contribution:** 2
**Rating:** 2
**Confidence:** 4

**Summary:**

The paper introduces a novel framework for training large language model (LLM) agents to engage in persuasive communication within social deduction games (SDGs) such as Werewolf, Avalon, and One Night Ultimate Werewolf (ONUW). The authors frame turn-based dialogue as a Stackelberg competition, where  the current player acts as the leader who strategically influences the follower’s response.

The authors propose an RL framework that fine-tunes LLM utterances to maximize persuasive impact, using Group Relative Policy Optimization (GRPO) rather than human feedback. They have collected a self-play dataset to refine base utterances into persuasive ones, and have measured persuasive impact by calculating how much an utterance shifts the probability distribution of the follower’s responses toward desired outcome.

**Strengths:**

- The paper treats persuasion as a leader-follower optimization problem, which is different from other approaches that typically concentrate on deducing other players’ roles and making strategic choices, which neglects the critical ability to influence other players’ beliefs and responses through persuasive communication.”
- They formalize dialogue as a Stackelberg game, and capture how each utterance can shape the trajectory of group reasoning
- The authors test their model on three distinct social deduction games (Werewolf, Avalon, ONUW), across multiple LLMs (GPT-4o, Gemini-2.5, Claude-3.5, GPT-5, Qwen3-14B)
- They show how integrated the Refiner with existing baselines consistently enhances performance, which leverages the strengths of established techniques while adding a persuasive dimension

**Weaknesses:**

- Given that the paper’s focus on developing AI agents capable of strategic social influence, I ask the authors to include some discussion of ethics, such as potential misuse (e.g., manipulative dialogue systems)
- The paper would benefit from including a human baseline, to understand the persuasive advantage of AI systems, and how your metric compares with human intuitive notions of deception
- The evaluation setting, although controlled, is quite narrow to study real persuasion. Specifically, the games that are studied have rigid rules, explicit goals, and a small number of agents. Whereas real-world persuasion such as negotiation, collaboration, or political discourse involves continuous conversation, partial observability, and long-term reputation, etc.

**Questions:**

- There are some missing related works such as [1]. There is also a new work [2] that proposes very similar objective to your paper. How does your work compare with [1] and [2]?
- The work uses win rate in social deduction games as the main metric for persuasive success, which may have several limitations. Are there other metrics the authors considered? This might be a useful paper to cite and compare against: https://arxiv.org/abs/2401.06373. Specifically, to understand and analyze what kind of persuasive strategies the model learns.

Related Works
[1] The MASK Benchmark: Disentangling Honesty From Accuracy in AI Systems. https://arxiv.org/abs/2503.03750
[2] Evaluating & Reducing Deceptive Dialogue From Language Models with Multi-turn RL: https://arxiv.org/abs/2510.14318

**Details Of Ethics Concerns:**

There is an ethics statement needed

---

### Note · Authors · 2025-11-24

**Comment:**

We sincerely appreciate the time and effort the reviewers dedicated to evaluating our work. After carefully considering, we have decided to withdraw the paper.

**Withdrawal Confirmation:**

I have read and agree with the venue's withdrawal policy on behalf of myself and my co-authors.